# New Markers of Disease Progression in Myelofibrosis

**DOI:** 10.3390/cancers13215324

**Published:** 2021-10-23

**Authors:** Rita Campanelli, Margherita Massa, Vittorio Rosti, Giovanni Barosi

**Affiliations:** 1Center for the Study of Myelofibrosis, General Medicine 2—Center for Systemic Amyloidosis and High-Complexity Diseases, IRCCS Policlinico San Matteo Foundation, 27100 Pavia, Italy; v.rosti@smatteo.pv.it (V.R.); g.barosi@smatteo.pv.it (G.B.); 2General Medicine 2—Center for Systemic Amyloidosis and High-Complexity Diseases, IRCCS Policlinico San Matteo Foundation, 27100 Pavia, Italy; m.massa@smatteo.pv.it

**Keywords:** chronic myeloproliferative neoplasms, primary myelofibrosis, inflammation, prognosis

## Abstract

**Simple Summary:**

Disease progression and prognosis in PMF are usually associated with worsening of anemia, increase of circulating blasts, and, more recently, with the presence, in addition to the “classical” driver mutations, of *JAK2*, *MPL,* and *CALR* genes, as well as of cytogenetic and molecular abnormalities that have been incorporated into new genetically based prognostic scoring systems. We have recently focused our attention on new biological markers of the disease, namely sIL-2Rα, eNAMPT, and CXCR4 expression on circulating CD34^+^ cells and CCL2 and VEGF-A polymorphisms, which have turned out to be associated with disease evolution and patient survival. Here, we discuss the role that recently described biological markers of the disease play in PMF progression and prognosis.

**Abstract:**

Primary myelofibrosis (PMF) is a myeloproliferative neoplasm due to the clonal proliferation of a hematopoietic stem cell. The vast majority of patients harbor a somatic gain of function mutation either of *JAK2* or *MPL* or *CALR* genes in their hematopoietic cells, resulting in the activation of the JAK/STAT pathway. Patients display variable clinical and laboratoristic features, including anemia, thrombocytopenia, splenomegaly, thrombotic complications, systemic symptoms, and curtailed survival due to infections, thrombo-hemorrhagic events, or progression to leukemic transformation. New drugs have been developed in the last decade for the treatment of PMF-associated symptoms; however, the only curative option is currently represented by allogeneic hematopoietic cell transplantation, which can only be offered to a small percentage of patients. Disease prognosis is based at diagnosis on the classical International Prognostic Scoring System (IPSS) and Dynamic-IPSS (during disease course), which comprehend clinical parameters; recently, new prognostic scoring systems, including genetic and molecular parameters, have been proposed as meaningful tools for a better patient stratification. Moreover, new biological markers predicting clinical evolution and patient survival have been associated with the disease. This review summarizes basic concepts of PMF pathogenesis, clinics, and therapy, focusing on classical prognostic scoring systems and new biological markers of the disease.

## 1. Primary Myelofibrosis: Disease Overview

### 1.1. Clinical Course, Diagnosis, Prognosis, and Treatment

Primary myelofibrosis (PMF) is a classical Philadelphia-negative chronic myeloproliferative neoplasm (Ph-neg MPN) of unknown etiology that originates from the clonal proliferation of a hematopoietic stem/progenitor cell [1].

It is characterized by variable degrees of bone marrow (BM) fibrosis, abnormal proliferation and trafficking of hematopoietic progenitor cells (HPCs), cytopenias that can affect one or more myeloid lineages, splenomegaly, increased risk of thrombotic events, and of blast transformation [2].

In 1951, thanks to the work of Sir William Dameshek, it was understood that PMF shares clinical and pathological features with essential thrombocythemia (ET) and polycythemia vera (PV), and that ET and PV can progress to myelofibrosis (MF) [3]: the recent discovery of a common mutational background has confirmed that these diseases represent different faces of the same disorder. In PMF, the median age at diagnosis is 67 years and the incidence rate varies from 0.8 to 2.1 per 10^5^ persons per year [4]. Disease evolution is characterized by cytopenia(s), in particular anemia, marrow failure, and thrombo-hemorrhagic complications [5,6], and 15–20% of patients progress toward a leukemic transformation [5]. Causes of death include leukemic transformation, cardiovascular events, and complications of cytopenias, including infections and bleeding.

Diagnosis of PMF relies on the histopathological analysis of the BM, which presents a variable degree of fibrosis. However, fibrosis is not essential for the diagnosis of the disease, as a pre-fibrotic condition (pre-MF) has been recognized by the World Health Organization (WHO) [2,7,8]. The BM pathognomonic histologic feature of PMF is the presence of an increased number of abnormal, tightly clustered, megakaryocytes that is often associated with an altered cellularity and an increased angiogenesis when compared to normal BM architecture [2,9,10].

Up to 90% of patients with PMF harbor at least one of these so-called “driver” mutations: V617F in the *JAK2* gene, W515L/K in the *MPL* gene, and 52 bp deletion (type 1 mutation) or 5 bp insertion (type 2 mutation) in the *CALR* gene [11]. Exome sequencing data have shown that additional exonic mutations in *ASXL1*, *DNMT3A*, *TET2*, *IDH1/2*, *DNMT3A*, *CBL*, *K-H-N-RAS*, *IKZF1*, *TP53*, *SF3B1*, *SRSF2*, and *U2AF1* can be detected [12,13,14].

Current practice utilizes for diagnosis the revised 2016 WHO diagnostic criteria for overt PMF and pre-MF encompassing three major and at least one minor criteria specific to histological, mutational, clinical, and laboratory-based features [2,15] (Figure 1).

The International Working Group Myeloproliferative Neoplasms Research and Treatment (IWG-MRT) diagnostic criteria are used for post-PV MF and post-ET MF [2,15,16] (Figure 2).

The prognosis of PMF is variable and the median survival is estimated around 6 years from diagnosis, ranging from 2 to 15 years, and in some cases even more [17]. The most frequently adopted prognostic models of PMF are the International Prognostic Scoring System (IPSS) [18], which is used at diagnosis, and a dynamic prognostic model (Dynamic International Prognostic Scoring System, DIPSS), which can be used at any time during the course of disease [19]. Both are based on clinical–laboratoristic parameters. The recent observation that the presence of additional mutations, besides driver mutations, independently impacts on patient prognosis, led to the incorporation of these additional mutations in new prognostic scores. The mutation-enhanced International Prognostic Scoring System (MIPSS70) and its updated version MIPSS70+ v2.0 were conceived for patients up to 70 years of age who were candidates for allogeneic hematopoietic stem cell transplantation (HSCT) [20]. At variance with MIPSS70 and MIPSS70+ v2.0, which include both clinical and genetic parameters, the genetically inspired prognostic scoring system (GIPSS) is exclusively based on genetics, in particular cytogenetics and mutational status [21]. A detailed description of these models and of their risk categories can be found in Tefferi et al. [22].

With regard to post-PV and post-ET MF, the myelofibrosis secondary to PV and ET (MYSEC) model, which is based on integrated clinical and molecular parameters, is widely used [23].

The treatment of PMF is often based on the clinical needs of patients in order to relieve disease-associated symptoms, prevent complications, and induce long-term complete remission [24,25]. Currently, the only recognized curative treatment for the disease is the allogeneic HSCT. However, this treatment can be offered to a minority of patients due to its intrinsic morbidity and mortality [26,27,28].

Current pharmacological treatments rely mainly on drugs such as hydroxyurea or JAK inhibitors such as Ruxolitinib or Fedratinib [29,30,31,32]. Unfortunately, patients undergoing treatment with Ruxolitinib lose their response to the drug or experience side effects that do not allow for the treatment to be continued at therapeutic doses [33]. Thus, other drugs or compounds have been investigated at experimental level and in clinical settings. They include others JAK inhibitors (pacritinib, momelotinib) [34,35], immunomodulatory imide drugs (IMiDs), and immunosuppressive and/or anti-inflammatory agents, including interferons and immune checkpoint inhibitors [36,37,38,39,40,41,42,43,44,45,46].

Other drugs, currently used only at experimental level in clinical trials, include Bromodomain and extra terminal protein (BET) inhibitors such as the following: CPI-0610 [47]; luspatercept, an activin receptor IIA ligand trap that improves anemia [48]; lPRM-151, a recombinant form of pentraxin 2, which has been shown to reverse fibrosis formation in preclinical models [49]; imetelstat, a potent telomerase inhibitor [50]; inhibitors of phosphatidylinositol 3-kinase (PI3K) [51]; inhibitors of the hedgehog pathway [52]; and inhibitors of histone deacetylase [53].

With the advent of JAK inhibitors, the occurrence of splenectomy in patients with myelofibrosis is rare. Current guidelines suggest that splenectomy should be restricted to selected patients with hemolysis, severe symptomatic splenomegaly, splenic infarction, portal hypertension due to splenomegaly, or severe hyper-catabolic symptoms that are refractory to non-surgical therapy [6].

### 1.2. Pathogenesis of PMF

In 2005, it was reported by different groups that about 60% of patients with PMF harbor an acquired gain of function mutation of the *JAK2* gene in their hematopoietic cells, resulting in a constitutive activation of the JAK/STAT signaling pathway [54,55,56,57,58,59]. This, in turn, results in the downstream activation of the proliferative signals that sustain the disease, at least at its onset and initial course. In the subsequent years, gain of function mutations in the *MPL* and in the *CALR* genes, also resulting in the JAK/STAT pathway activation, were identified [60,61,62,63,64,65,66,67]. About 10% of PMF patients do not harbor in their hematopoietic cells any of the three canonical driver mutations and are referred to as “triple negative”: nevertheless, an activated signaling of the JAK/STAT pathway has been documented in these patients, in some cases as a consequence of mutations affecting genes that negatively regulate JAK2 activity, such as LNK [68], whereas in others it is as a consequence of atypical mutations in the three driver genes [12].

Mutations in genes acting as epigenetic regulators of gene expression have been identified in a large number of PMF patients [69], These genes include epigenetic modifiers (*TET2*, *ASXL1*, *IDH1* and *2*, *EZH2*, and *DNMT3A*), regulators of RNA splicing (*SRSF2*, *U2AF1*, and *SF3B1*), tumor suppressors (*TP53*), and regulators of cytokine signaling (*CBL*). In particular, *ASXL1*, *SRSF2*, *EZH2*, *IDH1,* and *IDH2* have been associated with a worse prognosis [22]. In PMF patients, specific microRNA (miRNA) signatures have been described [70]. In particular, an altered miRNA expression has been described as involved in the megakaryocytic hyperplasia that characterizes PMF BM [71,72]. Thus, histone modifications, DNA methylation, and aberrant miRNA expression are thought to play a relevant role in PMF pathogenesis [73,74].

A disrupted crosstalk between hematopoietic stem cells (HSCs) and cells constitutive of the stem cell niche, consequent to an altered marrow and splenic microenvironment, has been recognized as a contributor to disease pathogenesis [75,76,77,78,79]. Studies in animal models indicate that these microenvironmental abnormalities may induce PMF independently from the presence of driver mutations [80]. For instance, Arranz et al. have shown that nestin-expressing mesenchymal stem cells (MSCs) are reduced in the BM of both MPN patients and of MPN mice models compared to their normal counterparts [81]. This reduction is triggered by IL-1β produced by mutant HSCs and favors mutant HSC expansion. We found that BM- and spleen-derived MSCs from patients with PMF harbor genetic abnormalities and display an altered functional activity, suggesting that a primary MSC defect may either lead to or favor the pathogenesis of PMF [82,83]. Moreover, we [84] and others [85], showed that some endothelial cells (ECs) from both the spleen and splenic vein of patients with PMF harbor the JAK2V617F mutation, suggesting that these mutant ECs could promote malignant cell expansion [86] and contribute to disease-related clinical manifestations [87].

These data support the hypothesis that, beside an intrinsically mutated HSC, an altered hematopoietic niche sustains the development of PMF in keeping with the hypothesis of the “bad seed in a bad soil” [76].

Chronic inflammation could also be involved in determining genetic damage to HSCs, as suggested by Hasselbalch through oxidative damage to DNA mediated by the accumulation of reactive oxygen species (ROS) [88,89], as well as determining the fibrotic process.

Recent data show that the presence of a chronic condition of inflammation provides a continuous stimulus for Gli1+ myofibroblasts to nurture the process of BM fibrosis [90]. It is likely that other MSC-derived myofibroblasts participate in collagen deposition in the BM of PMF, contributing to fibrosis [91]. For instance, leptin receptor-expressing stromal cells can differentiate into myofibroblasts that, through a PDGFRα/β-dependent signaling, are involved in reticulin deposition in PMF BM [92].

Thus, PMF pathogenesis seems to be sustained by a multi-factorial mechanism, where constitutive JAK/STAT activation, epigenetic dysregulation, and an altered BM and splenic microenvironment interact together, at different extent, in determining the disease phenotype.

## 2. New Markers of Disease

C-reactive protein (CRP) is an acute-phase reactant protein used in clinics as a marker of inflammation, triggered by pro-inflammatory cytokines such as IL-6, IL-1β, and TNF-α.

In 2013, Barbui and colleagues [93] reported in patients with PMF that the blood levels of high-sensitivity (hs)-CRP were significantly higher than in age-matched healthy subjects. Patients with the highest hs-CRP concentration were older, homozygous for the JAK2V617F mutation, and mostly in the high-risk prognostic group. hs-CRP has also been significantly associated with relevant thrombotic events in patients with ET and PV [93].

Studying the possible causes of inflammation in PMF [94], our group confirmed high plasmatic hs-CRP levels in a huge cohort of selected patients, out of therapy and not suffering from acute inflammatory, autoimmune diseases, or other neoplasms. hs-CRP levels were related to disease progression and a higher risk of blast transformation and worse survival, confirming the data obtained by Barbui and colleagues [93]. Besides, plasmatic concentration of hs-CRP was significantly associated with older age, JAK2V617F allele burden ≥ 50%, advanced BM fibrosis, and an increased percentage of circulating CD34+ cells, blasts, and increased absolute monocyte count. Altogether, the data suggest an association among inflammation and a phenotype of disease progression, a high risk of blast transformation, and worse survival [94]. Thus, hs-CRP has emerged as a useful prognostic factor in MPNs, and its role should be further investigated.

The soluble form of IL-2Rα (sIL-2Rα) is generated by the proteolytic cleavage of the membrane IL-2Rα on activated T lymphocytes and exerts immunoregulatory effects controlling the T cell responses [95]. Elevated concentrations of sIL-2Rα have been observed in cancers and autoimmune, inflammatory, and infectious diseases [96,97]. In MPNs, sIL-2Rα was shown to correlate with disease progression, poorer survival, and blast transformation-free survival [98].

In 2020, our group [99] demonstrated that sIL-2Rα was significantly higher in PMF patients (one-third at diagnosis) than in healthy subjects and that the plasmatic concentration of sIL-2Rα was directly correlated with various markers of disease progression (number of circulating CD34+ cells and blasts, decreased hemoglobin and platelet concentration, and increased spleen size). Furthermore, the analysis of newly diagnosed PMF indicated a correlation between sIL-2Rα levels in plasma and disease progression in JAK2V617F-mutated patients, but not in those with the CALR mutation [99]. The mechanism underlying this phenomenon remains unexplained, nevertheless suggesting a different pathogenetic mechanism between JAK2V617F- and CALR-mutated patients that could influence the therapeutic approach.

The intracellular enzyme nicotinamide phosphoribosyltransferase (NAMPT) converts nicotinamide into nicotinamide mononucleotide, which is fundamental for cellular metabolism, energy production, DNA repair, and survival [100]. NAMPT also exists as an extracellular protein (eNAMPT), released from different types of cells, defined as Pre-B cell colony-enhancing factor or as an adipokine called visfatin [101].

eNAMPT is involved in the pathogenesis of various human diseases with an inflammatory basis (i.e., rheumatoid arthritis, type 2 diabetes, sepsis, and tumorigenesis) [102,103]; besides, the expression of NAMPT in tissues was found upregulated in cancers and hematologic malignancies [104].

In a paper published by our group [105], PMF patients were enrolled to verify the association between plasmatic eNAMPT and the phenotype/progression of the disease. In patients with PMF, eNAMPT was fivefold higher than in controls and higher eNAMPT was associated with increasing hemoglobin, leukocytes, and platelet count. The data evidenced no correlation between eNAMPT levels and disease duration, spleen size, circulating CD34+ cells, and cholesterol level in serum; moreover, there was no difference between eNAMPT concentration and different genotypes. Results indicate that eNAMPT levels were increased in a huge number of PMF patients and higher levels of eNAMPT mark the disease hyperproliferative potential. Our group [105] interpreted the association between eNAMPT and the increased number of mature blood cells, hypothesizing that eNAMPT could be essential for myeloproliferation and an important permissive agent for the malignant hematopoietic proliferation and/or differentiation. Moreover, elevated levels of eNAMPT in patients with PMF predicted a slowing of the disease progression toward a worsening phenotype and blast transformation, whereas lower levels were correlated with a briefer interval to blast transformation. Starting from the assumption that eNAMPT has been involved in the regulation of inflammatory responses [106], the hypothesis that eNAMPT behaves as an inflammatory cytokine in PMF has yet to be proven, considering the lack of a significant correlation between eNAMPT and plasmatic hs-CRP.

The changes in the eNAMPT concentration induce the activation of HPCs/HSCs mobilization, dependent on the chemokine (C-X-C Motif) receptor 4 (CXCR4). The mechanism is NAMPT-induced NAD-dependent Sirtuin1 activity, which leads to a reduced CXCR4 expression on hematopoietic cells in PMF, caused by a hyper-methylation of the *CXCR4* gene promoter [107,108,109]. In 2007, we reported that the low membrane expression of CXCR4 on circulating CD34+ in PMF patients was correlated with the mobilization of cells [107].

It is well known that the interaction between CXCR4 and its ligand, the chemokine CXCL12 (also known as SDF-1), plays a pivotal role in cell mobilization. Cho et al. [110] reported that, although PMF patients can display elevated levels of immunoreactive forms of CXCL12 both in peripheral blood (PB) and BM, these are represented mainly by truncated proteins, never detectable in the PB and BM of healthy subjects, which lack any activity as chemoattractant for CD34+ cells [110]. Thus, both a reduced expression of CXCR4 and a not effective form of CXCL12 can be responsible for the high mobilization of CD34+ cells that is observed in PMF patients. More recently, it has been suggested that JAK2 activation in PMF patients can activate, through PI3K signaling, the CXCL12/CXCR4 pathway, adding a new potential mechanism to CD34+ cell trafficking in PMF patients [111].

Since the mobilization of CD34+ cells correlates with disease activity [112], we hypothesized that CXCR4 surface expression on CD34+ cells might characterize the disease progression in MPNs. In 2020, we evaluated circulating CD34+CXCR4+ in a huge cohort of MPNs and demonstrated that patients with PMF are characterized by a significantly reduced expression of CXCR4 on CD34+ cells where it identifies a myelodepletive phenotype [113]. In PMF, male gender, older age, and MPL mutation were independently correlated with reduced CD34+CXCR4+ cells and associated with a briefer interval to develop severe anemia, large splenomegaly, thrombocytopenia, leukopenia, elevated CD34+ blood cells, blast transformation, and death. The data allowed us to construct a prognostic model of survival using CD34+CXCR4+ < 39% as a cut-off point at diagnosis. Other co-variates comprised the following: age > 65 years, hemoglobin < 10 g/dL, and CD34+ cells > 50 × 10^6^/L. CXCR4 expression on circulating CD34+ cells seems to be a sensitive marker of disease activity and a new potential diagnostic and prognostic biomarker in PMF. In conclusion, we found that, in PMF patients, the CXCR4 expression on CD34+ cells was strongly correlated with severe anemia, thrombocytopenia, leukopenia, increased concentration of blood CD34+ cells, larger spleen size, and severe BM fibrosis.

In agreement with the notion of an elevated number of circulating CD34+ progenitor cells is the finding that a high number of colony-forming cells can be detected in the blood of PMF patients by in vitro semisolid growth assay [114,115]. More recently, it was shown in 110 PMF patients that the number of hematopoietic colonies (both CFU-GM and BFU-E) above the 75th percentile was an independent adverse prognostic factor [116]. In keeping with this observation, Geissler et al., by using semisolid in vitro cultures, reported that a skewed differentiation toward the myelomonocytic lineage compared that toward the erythroid lineage was associated to higher leukocyte count, higher blast frequency, lower hemoglobin and platelet levels, higher frequency of additional mutations, and shorter survival. Thus, by assessing the degree of myelomonocytic skewing in in vitro cultures, it was possible to discriminate a subgroup of PMF patients with the worst prognosis compared to PMF patients not showing skewed myelomonocytic differentiation [117].

Single nucleotide polymorphisms (SNPs) are important players in modulating the individual pro-inflammatory background [118]. SNPs that affect cytokine/chemokine gene expression seem to have a role in diseases with an inflammatory component such as PMF, both modifying the individual pro-inflammatory background and having implications in the MPN setting, characterized by aberrant cytokine production.

Chemokine (C-C motif) ligand 2 (CCL2) is a member of the C-C class of the β chemokine family and one of the most potent immunomodulatory chemokines known to be elevated in PMF [119].

In a work by Masselli et al. [120], the authors demonstrated that an A to G substitution in the distal regulatory region of the *CCL2* gene at position −2518 from the transcription start site (rs1024611 SNP) influences the transcriptional activity of *CCL2* due to a mechanism of allelic expression imbalance and preferential transcription of the G allele. The functional effect of the rs1024611 SNP on CCL2 expression is dose dependent, with cells from homozygous individuals producing more chemokines than cells from heterozygous individuals [121]. The homozygosity for the rs1024611 SNP of the chemokine CCL2 in PMF, particularly in male patients, represents a high-risk variant and a novel host genetic determinant of reduced overall survival (OS). These results provide opportunities for CCL2 SNP genotyping as a potential novel strategy to risk stratify patients. Besides, the results provide a novel mechanism underlying the anti-inflammatory effects of Ruxolitinib, via the simultaneous downregulation of CCL2 production and CCR2 expression in PMF cells [120].

In the last year, our group analyzed three different polymorphisms of the vascular endothelial growth factor A (*VEGF-A*) gene (rs2010963, rs3025020, and rs3025039) to evaluate their possible role in PMF development and prognosis.

VEGF-A is a proangiogenic protein correlated with development and progression of myeloproliferative neoplasms [122] and it is well known that SNPs in the *VEGF-A* gene are associated with incidence and prognosis of many solid and hematologic cancers [123].

We reported that VEGF-A SNP + 405 G>C (rs2010963) genotypes correlate with a pre-MF phenotype with risk of thrombosis, especially deep vein thrombosis in atypical sites, in addition with vascular complications. These results could indicate that in PMF a prefibrotic phenotype at diagnosis is influenced by germ-line genetic factors and could explain the high incidence of thrombosis in pre-MF [8], suggesting converging and independent causes for pre-MF and thrombosis [124].

Another study conducted by our group [125] indicated that CALR-mutated PMF patients, especially type-2/type-2-like patients, have an increased frequency of VEGF-A rs3025020 minor allele CT/TT genotypes. PMF patients with a rs3025020 T-allele genotype had a lower cumulative incidence of thrombosis, particularly deep vein thrombosis. These data indicate that patients with PMF and VEGF-A rs3025020 minor T-allele genotypes are more likely to have a CALR driver mutation, in comparison with other driver mutations, and a lower incidence and hazard for deep vein thrombosis [125].

Single nucleotide variants (SNVs) can affect VEGF-A gene expression, altering key regulatory sequences or mRNA stability at key regulator loci.

Among the SNVs identified within VEGF-A, the +936C > T polymorphism (rs3025039) located in 3′ UTR of the gene has been associated with various diseases, including cancers [126].

We showed that PMF patients carrying the rs3025039 minor T-allele displayed increased predisposition to acquire the JAK2V617F driver mutation. In JAK2-mutated patients, the TT genotype was associated with more severe disease at diagnosis, i.e., reduced platelet count, increased lactate dehydrogenase in plasma, massive splenomegaly, high number of CD34+ cells in blood, lower cholesterol concentration, and a higher inflammatory background as indicated by the hs-CRP-elevated plasmatic level. In accordance with these data, these patients also displayed a higher risk to develop thrombocytopenia during the disease evolution. More interestingly, the association with disease severity of the SNV translates into a reduced OS in JAK2V617F-mutated individuals [127].

All these findings help to improve our knowledge about the genetic basis of PMF and, if confirmed, they could have important clinical and prognostic implications.

Besides the new markers described above, new molecules and cell subsets under current investigation could be associated with the disease course and prognosis in the future.

ROS are oxygen-containing molecules involved in many biological processes including cellular signaling, immune defense, and several inflammation-driven diseases. An elevated ROS production is thought to have a role in tissue damage, dysfunction, and fibrosis. The excessive ROS production in MPN patients induces a proliferative advantage of JAK2-positive clones [128,129].

As previously described, inflammation is a pathobiological feature of PMF and various evidence indicates that the phenotype of the disease is strongly influenced by inflammatory and immune mechanisms [130].

Furthermore, Zhang et al. reported in a mouse model that the decreased expression of CXCR4 correlates with increased endogenous ROS levels, which may cause genomic instability subverting both DNA damage and repair response [131]. Koschmieder and colleagues [132] also demonstrated that ROS, major players in inflammation-induced oxidative damage to cellular components, have a relevant role in the pathogenesis of MPNs, where the malignant clone itself produces an excess of ROS, thereby creating a self-perpetuating circle in which ROS activate proinflammatory pathways, which in turn create more ROS. Interestingly, it has been shown in animal models that the presence of the JAK2V617F mutation induces the accumulation of ROS in HSCs, increases the DNA damage, and favors the development of myeloproliferation [133]. The oxidative stress is a mechanism of PMF disease progression that is specifically dependent on inflammation, and ROS accumulation has a role in determining the damage to the HSC (creating a high-risk microenvironment in the hematopoietic niche), and the reduction of the cellular oxidative stress could weaken stem cell proliferation. The identification of a category of PMF patients with elevated levels of inflammation, older age, and high JAK2V617 allele burden could allow us to target the accumulation of ROS and accordingly prevent or counteract the progression of the disease.

Even if suppressive myeloid cells were described more than 30 years ago in cancer [134], the functional importance of myeloid-derived suppressor cells (MDSCs) in the immune system has only recently been appreciated. MDSCs are heterogeneous cell populations of myeloid origin that accumulate in patients with cancer, sepsis, or chronic inflammation. In the steady state, immature myeloid cells are present in the BM and migrate to different peripheral organs, where they differentiate into macrophages, dendritic cells, or granulocytes. Cytokines and chemokines produced during acute/chronic infections, trauma, or sepsis and in the tumor microenvironment recruit immature myeloid cells in the PB, spleen, liver and lymphoid organs, prevent their differentiation, and induce their activation [135].

MDSCs could play a role in PMF disease progression by inducing oxidative stress, inhibiting the function of immune cells, and having a direct role in the abnormal angiogenesis that characterizes PMF.

In 2016, Wang et al. [136] found that MDSC levels were increased in the PB of MPN patients when compared to healthy subjects, but these levels did not correlate with spleen size, leukocytes, platelet count, hemoglobin levels, or JAK2V617F allele burden, probably due to the small number of patients analyzed. MDSCs could play an important role in the chronic inflammatory status that characterizes PMF: (1) they are probably one of the cell population responsible for the production of ROS, the mechanism of oxidative stress by which inflammation induces disease progression; (2) the correlation of MDSCs with clinical parameters could shed light to the different outcomes and responses to therapy that characterizes patients, also according to the mutational status. Moreover, a detailed characterization of the frequency and function of the different MDSC subpopulations could be useful, particularly if it is correlated to the inflammatory status or disease progression of patients. Besides, immunotherapy with MDSC-related inhibitors of differentiation remains to be explored.

## 3. Conclusions

Routinely used prognostic scoring models are currently based on clinical symptoms, laboratoristic parameters, and genetic and molecular markers. This has allowed medical professionals to define risk categories that have become progressively more precise. Nonetheless, the identification of new markers of disease predicting clinical progression and patient survival in a more refined prognostic models will offer the possibility to improve our capacity of patient stratification, resulting in more tailored and efficient therapeutic approaches.

## Figures and Tables

**Figure 1 cancers-13-05324-f001:**
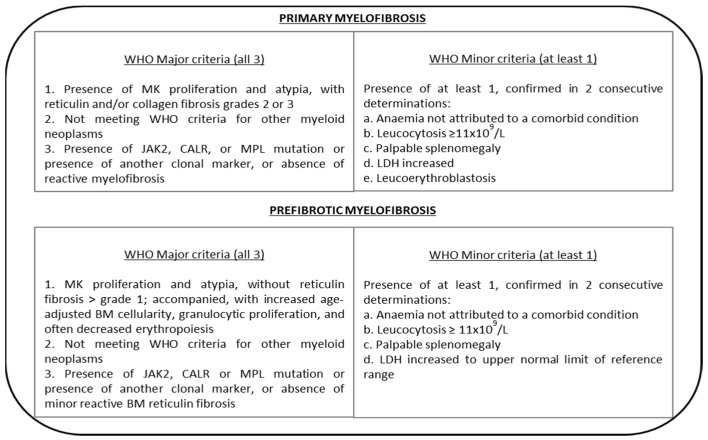
Summary of diagnostic WHO criteria for primary myelofibrosis (PMF) and prefibrotic myelofibrosis (pre-MF). Abbreviations: MK = megakaryocyte; LDH = lactate dehydrogenase; WHO = World Health Organization. Adapted from Arber et al. Blood 2016 [2] and Gowin et al. Leuk Res 2015 [15].

**Figure 2 cancers-13-05324-f002:**
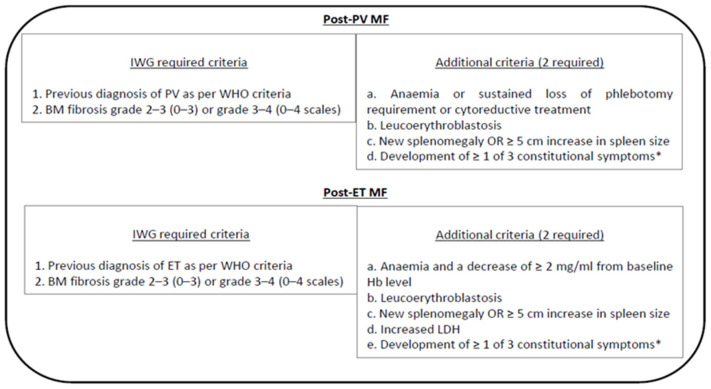
Summary of IWG diagnostic criteria for post polycythaemia myelofibrosis (post-PV MF) and post essential thrombocythemia myelofibrosis MF (post-ET MF). Abbreviations: Hb = hemoglobin; LDH = lactate dehydrogenase; PV = polycythemia vera; ET = essential thrombocythemia; BM = bone marrow; IWG = International Working Group. * Constitutional symptoms: >10% weight loss in 6 months, night sweats, and unexplained fever (>37.5 °C). Adapted from Arber et al. Blood 2016 [2] and Gowin et al. Leuk Res 2015 [15].

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
