# Peer review of "New Markers of Disease Progression in Myelofibrosis"

_cancers, 2021, doi:10.3390/cancers13215324_

Round 1

Reviewer 1 Report

The manuscript is well written.

CXCL12 is well known ligand for CXCR4. Authors should discuss the association between the ligand and receptor in patients with MF. 

Author Response

CXCL12 is well known ligand for CXCR4. Authors should discuss association between the ligand and receptor in patients with MF

We thank the reviewer for the suggestion to examine the role of CXCL12 and CXCR4 as a mechanism of stem cells mobilization in PMF. We have now discussed in paragraph “New markers of disease” the interaction between CXCR4 and its ligand CXCL12 as a mechanism underlying the high CD34+ cell trafficking in peripheral blood of PMF.

Reviewer 2 Report

Optimizing treatment concepts in cancer requires deeper and more comprehensive understanding of the pathophysiology of the disease. Therefore, a review like this summarizing new markers of disease progression in myelofibrosis is certainly of interest for physicians who manage patients with MPN. The authors describe in a well structured and comprehensive way what findings may help to better understand disease biology and their potential clinical impact. Some findings that may be also important in this respect, however,  have not been  mentioned and should be included in this list. Particularly the in vitro characterization of MPN by semisolid cultures has a long tradition and may complement the findings regarding the CD34+ cell data that have been presented by the authors.

Comments

It is known that patients wit MF have a high number of circulating progenitor cells as measured by semisolid cultures (Chervenick PA, Blood 1973) and by the determination of CD34+ cells. In this respect, the finding of reduced expression of CXCR4 on MF progenitor cells is highly interesting and may provide one possible explantation for this phenomenon considering the critical role of CXCR4 in progenitor cell mobilization. This should be discussed. Moreover, the number of circulating progenitor cells, as measured by in vitro cultures, have been shown to have a significant impact on OS in patients with MF (Sagaster V et al, Haematologica 2003). Furthermore, myelomonocytic skewing in semisolid cultures was also a major prognostic parameter in MF patients (Geissler K et al, Cancers 2020). Moreover, in this paper it was shown that myelomonocytic skewing in these patients was associated with a higher number of mutations. Although semisolid in vitro cultures are not routinely performed in the diagnostic work up of MPN patients, and are therefore not good candidates for prognostic scoring systems, these findings may provide a better understanding of disease biology.

Author Response

It is known that patients wit MF have a high number of circulating progenitor cells as measured by semisolid cultures (Chervenick PA, Blood 1973) and by the determination of CD34+ cells. In this respect, the finding of reduced expression of CXCR4 on MF progenitor cells is highly interesting and may provide one possible explantation for this phenomenon considering the critical role of CXCR4 in progenitor cell mobilization. This should be discussed. Moreover, the number of circulating progenitor cells, as measured by in vitro cultures, have been shown to have a significant impact on OS in patients with MF (Sagaster V et al, Haematologica 2003). Furthermore, myelomonocytic skewing in semisolid cultures was also a major prognostic parameter in MF patients (Geissler K et al, Cancers 2020).”

We thank the reviewer for her/his relevant comment. Following her/his suggestions, we have now discussed the role of CXCR4 and its ligand CXCL12 in stem cells mobilization in PMF in the paragraph “New markers of disease”. We also have quoted and commented the manuscript by Sagaster V. et al. (Haematologica 2003) and Gessler K. et al. (Cancers 2020) highlighting the number of circulating progenitor cells measured as hematopoietic colonies in in vitro culture and its stewing toward the myelomonocytic lineage as a major prognostic factor in PMF patients. We agree that in vitro cultures are not easily comparable among different laboratories and therefore cannot be added as a routine test in the designing of a prognostic scoring system. However, they do really provide important information for a deeper understanding of the biology of the disease.

Reviewer 3 Report

The manuscript by Campanelli et al. is intended to describe “new markers of disease progression in myelofibrosis”. The expectations deriving from the attractive title are actually disappointed by the reading of the text, which sounds more like a gross wordy review on myelofibrosis rather than an up-to-date focus on new biomarkers in MF and, in particular, on their clinical impact. To be considered for publication, the manuscript needs to be massively rearranged, by balancing the introduction (now accounting for 2/3 of the paper and, more important, completely pointless in the current form) with the planned focus (i.e., new markers). More in details, paragraphs 1 and 2 should be summarized into a brief pointed introductive section (that should be shorter than the section/s dedicated to the new biomarkers), possibly referring to comprehensive up-to-date reviews [e.g., for diagnostics and risk stratification: Tefferi Primary myelofibrosis: 2021 update on diagnosis, risk-stratification and management (already cited), Nann & Fend Synoptic Diagnostics of Myeloproliferative Neoplasms: Morphology and Molecular Genetics Cancers 2021; for genetics: Greenfield et al. Molecular pathogenesis of the myeloproliferative neoplasms J Hematol Oncol 2021; for microenvironmental changes: Nasillo et al. Inflammatory Microenvironment and Specific T Cells in Myeloproliferative Neoplasms: Immunopathogenesis and Novel Immunotherapies Int J Mol Sci 2021], in order to avoid redundant digressions on issues already extensively reviewed elsewhere, thus facilitating the global readability of the paper. In particular, by considering the focus of the review, (i) the tables detailing the WHO diagnostic criteria (ii) the redundant lists of signs/symptoms, as well as (iii) the point-by-point lists of each single parameter affecting prognosis in each single prognostic scoring system (with all the related survivals), basically appear unnecessary and aimless. Of note, same concepts about hs-CRP are reported both in subparagraph 2.2. (microenvironment) and paragraph 3 (new markers), for a total of 26 lines dedicated to hs-CRP going around two references, thereby further adding redundancy. Moreover, the paragraph 4 (“perspectives”) seems just a prosecution of paragraph 3, rather than a conceptually autonomous section. 

Globally, the quality of the manuscript is low, being heavily undermined by several conceptual and formal errors, as well as by a general lack of accuracy. By way of example (but not limited to):

  • In the abstract, cytopenias and splenomegaly are referred to as “symptoms”, rather than clinical-laboratoristic features
  • “PV, ET and PMF shared common clinical-pathological features and could transform into each other [REF 3: Dameshek, 1951]”. Are the authors sure of this assumption? It seems that PMF can transform into ET or PV...
  • “PV is characterized by an excessive proliferation of the erythroid lineage”. PV is rather characterized by a trilinear hyperplasia (aka panmyelosis).
  • Authors: “In PMF […] the incidence rate varies from 0.8 to 2.1 per 10persons per year [6]”. Reference number 6 [Thapa et al. 2021]: “The incidence rate varies from 0.8 to 2.1/100,000/year”. Please note that 100,000 = 105.
  • “Diagnosis of PMF relays on the histopathological analysis of the bone marrow (BM) which often presents a variable degree of fibrosis (caused by an excess of deposition of collagen fibers)”. Please note that collagen fibers are detected only in late-stage fibrosis (see Thiele et al. Haematologica 2005; Bauermeister et al. Am J Clin Pathol 1971), while in early phases fibrosis is sustained by reticulin fibers.
  • “The most frequently adopted prognostic modeling […] leukocytes 25 x 109/L” (more than, less than or equal to?)
  • Several syntactic and grammatical errors occur throughout the paper. Just to mention some: “patognomic” (did you mean “pathognomic” or, better, “pathognomonic”?); “relays” occurs three times throughout the paper (did you mean “relies”?); “revisied” (did you mean “revised”?); in general, the use of singular vs plural needs to be accurately checked throughout the paper (see for examples: “clinical and histological feature”, “platelet”, “osteoblast”, etc…); likewise, the use of “which”/”that” and “among”/”between” needs to be proofread; moreover “may complicates”, etc.
  • The use of abbreviations/acronyms is totally anarchic: some are explained (e.g., VEGF, PD-1, CTLA-4), whereas others are not (e.g., TNF, FGF, TGF, ECs…). CRP is explained twice. Some others (such as HCT) are not explained the first time they occur in text but only thereafter; “hematopoietic stem cell transplantation" should be HSCT instead of HCT. Both “BM” and “bone marrow” discretionally occur. Lack of uniformity for indicating the same parameter or unit of measure (WBC/leukocytes, hemoglobin/haemoglobin/Hb, PD1/PD-1, 10 g/dL/100 g/L, dl/dL)
  • Lack of catching focused images/tables (other than copy-pasted diagnostic criteria)

Author Response

To be considered for publication, the manuscript needs to be massively rearranged, by balancing the introduction (now accounting for 2/3 of the paper and, more important, completely pointless in the current form) with the planned focus (i.e., new markers). More in details, paragraphs 1 and 2 should be summarized into a brief pointed introductive section (that should be shorter than the section/s dedicated to the new biomarkers), possibly referring to comprehensive up-to-date reviews

We agree with the reviewer on the necessity of balancing the introduction with the “new markers” section. According to the reviewer suggestion, we have rearranged paragraphs 1 and 2 summarizing them in a “disease overview” section. In its current revised form the introduction accounts for 1558 words whereas the new markers section accounts for 2607.

In particular, by considering the focus of the review, (i) the tables detailing the WHO diagnostic criteria (ii) the redundant lists of signs/symptoms, as well as (iii) the point-by-point lists of each single parameter affecting prognosis in each single prognostic scoring system (with all the related survivals), basically appear unnecessary and aimless.”

We have now eliminated both the redundant list of signs/symptoms and the list of parameters affecting prognosis including the related survivals. We have not deleted the tables of the WHO diagnostic criteria since we believe it can be of interest for those readers who are not confident with PMF and its prefibrotic and overt forms.

Of note, same concepts about hs-CRP are reported both in subparagraph 2.2. (microenvironment) and paragraph 3 (new markers), for a total of 26 lines dedicated to hs-CRP going around two references, thereby further adding redundancy.”

The reviewer is definitively right and in the revised version we have corrected this redundancy by eliminating the part in subparagraph 2.2.

“The paragraph 4 (“perspectives”) seems just a prosecution of paragraph 3, rather than a conceptually autonomous section.

According to reviewer suggestion, paragraph 4 has been merged with former paragraph 3 (now paragraph 2).

In the abstract, cytopenias and splenomegaly are referred to as “symptoms”, rather than clinical-laboratoristic features

This has been corrected according to reviewer suggestion.

PV, ET and PMF shared common clinical-pathological features and could transform into each other [REF 3: Dameshek, 1951]”. Are the authors sure of this assumption? It seems that PMF can transform into ET or PV...”

We apologize for the wrong sentence that has now been deleted and replaced properly.

PV is characterized by an excessive proliferation of the erythroid lineage”. PV is rather characterized by a trilinear hyperplasia (aka panmyelosis)”

We agree with the reviewer, this sentence is not anymore present in the manuscript.

In PMF […] the incidence rate varies from 0.8 to 2.1 per 10persons per year [6]”. Reference number 6 [Thapa et al. 2021]”

We apologize for this mistake that now has been corrected.

Please note that collagen fibers are detected only in late-stage fibrosis (see Thiele et al. Haematologica 2005; Bauermeister et al. Am J Clin Pathol 1971), while in early phases fibrosis is sustained by reticulin fibers.”

The reviewer is right and this sentence is not anymore present in the revised manuscript.

The most frequently adopted prognostic modeling […] leukocytes 25 x 109/L” (more than, less than or equal to?)”

We apologize: it was supposed to be >25 x 109/L. In the revised version this sentence doesn’t appear anymore.

Several syntactic and grammatical errors occur throughout the paper. Just to mention some….”

The reviewer is definitely right regarding the syntactic and grammatical errors and we do apologize about that. We thoroughly revised the entire manuscript to correct them.

The use of abbreviations/acronyms is totally anarchic….”

Thank you for this helpful criticism. In this revised version we have explained the abbreviations used in the text and we have adopted an uniform style for indicating the same parameters or unit of measure.

Round 2

Reviewer 3 Report

Well done! I really appreciated the modifications made by the Authors, as well as the point-by-point reply. The quality and readability of the paper are remarkably improved in the present version. However, as previously suggested, please consider to add some up-to-date references (of note, only 7/90 references cited in the introduction have been published in the last 2 years and less than 1/3 in the last 5 years), in order to offer the broad readership, part of whom may be not confident with PMF, the chance to deepen some relevant specific topics, comprehensively reviewed elsewhere. In particular:

  • Molecular pathogenesis of the myeloproliferative neoplasmsJ Hematol Oncol. 2021 (doi.org/10.1186/s13045-021-01116-z) for genetics
  • Inflammatory Microenvironment and Specific T Cells in Myeloproliferative Neoplasms: Immunopathogenesis and Novel Immunotherapies. Int J Mol Sci. 2021 (doi.org/10.3390/ijms22041906) for microenvironmental, inflammatory and immunological changes.

This latter could also fit the following sentence: "As previously described, inflammation is a pathobiological feature of PMF and various evidence indicate that the phenotype of the disease is strongly influenced by inflammatory and immune mechanisms", currently lacking a reference.

Author Response

“However please consider to add some up-to-date references (of note, only 7/90 references cited in the introduction have been published in the last 2 years and less than 1/3 in the last 5 years), in order to offer the broad readership, part of whom may be not confident with PMF, the chance to deepen some relevant specific topics, comprehensively reviewed elsewhere.”

Thank you for this helpful criticism; we agree with the reviewer on the necessity of adding some up-to-date references.

According to her/his suggestion we have included the papers by Greenfield G. et al. (J Hematol Oncol 2021) and Nasillo V. et al (Int J Mol Med Sci. 2021) in the revised version of the manuscript.